# Generation of a New Glutinous Photothermosensitive Genic-Male-Sterile (PTGMS) Line by CRISPR/Cas9-Directed Mutagenesis of Wx in Rice (*Oryza sativa* L.)

Kaichong Teng [1,2], Xin Wang [1,2], Xinying Guo [1,2], Yaoguang Liu [3,*] and Rongbai Li [1,2,*]

1   College of Agriculture, Guangxi University, Nanning 530004, China; 13471035890@163.com (K.T.); xinwang0112@126.com (X.W.); guoxinying24@126.com (X.G.)
2   State Key Laboratory for Conservation and Utilization of Subtropical Agri-Bioresources, Guangxi University, Nanning 530004, China
3   State Key Laboratory for Conservation and Utilization of Subtropical Agricultural Bioresources, South China Agricultural University, Guangzhou 510642, China
*   Correspondence: ygliu@scau.edu.cn (Y.L.); lirongbai@126.com (R.L.)

**Abstract:** The Photothermosensitive Genic-Male-Sterile (PTGMS) line, Y58S, an indica rice variety, combines high-quality and high-light-efficiency use, disease and stress resistance, and excellent plant type and mating force. Y58S is widely used to assemble two-line hybrid rice varieties, especially super hybrids. The *Wx* gene is the main effector gene for controlling amylose synthesis, which determines the amylose content (AC) of rice grains. By editing this gene, a glutinous line with a low AC can be obtained. In this study, the CRISPR/Cas9 system was used to mediate the editing of the *Wx* gene, which caused ultra-low AC mutations that produced a PTGMS glutinous rice strain with excellent waxiness. The results showed that 18 positively transformed plants were obtained from the $T_0$ generation, with a mutation rate of 64.29%, of which six were homozygous mutant plants, indicating that the gene-editing target had a higher targeting efficiency and a higher homozygosity mutation rate. Compared to the wild type, the AC of the mutants was significantly lower. Through molecular marker detection and screening of $T_1$ and $T_2$ generations, five homozygous T-DNA-free mutant strains were identified that were consistent with Y58S in fertility and other agronomic traits except for AC. Among these, the AC of the W-1-B-5 homozygous mutant, the glutinous PTGMS line wx-Y58S, was as low as 0.6%. Our research revealed that the *Wx* gene of excellent PTGMS rice can be edited to generate a new waxy PTGMS line using the CRISPR/Cas9 system. This study provided a simple and effective strategy for breeding high-yield, high-quality, and glutinous two-line hybrid rice, and provided excellent sterile lines for their large-scale application. Once put into use, waxy hybrid rice will greatly improve the yield of glutinous rice and increase social benefits.

**Keywords:** rice; genome editing; crispr/cas9; waxy; amylose content (ac); y58s

## 1. Introduction

More than half of the world's population depends on rice as a dietary staple, particularly in China. However, it is no longer associated just with satiety, but rather with a better quality of life. Rice production is integral to China's national development, and meeting the demand for high-quality rice is urgent [1], which includes more refined processing and adapting to the tastes of people in different regions [2].

Starch, the main component of rice endosperm, accounts for about 75% of its quality. According to its chemical structure, rice starch can be divided into two types: amylose and amylopectin. Amylose content (AC) is an important index for evaluating rice quality, which determines the taste and cooking properties of rice [3]. Amylose is a high molecular polymer with few branches connected by 1–4 glycosidic bonds. The molecular weight is 105–106 Da and combines easily with iodine to produce a dark-blue color. On the other

hand, amylopectin has many branches. Like amylose, it has 1–4 glycosidic bonds as the main chain, but dextran branches are generated at the 1–6 glycosidic bonds. The molecular weight is 107–108 Da. It does not easily combine with iodine and appears red-brown [4].

The AC directly determines the cooking characteristics, processing, and testing of rice [5], and according to the AC, rice varieties can be divided into five categories—waxy ($\leq$2%), very low (2.1−9%), Low (9.1−20%), medium (20.1–25%), high ($\geq$25.1%)—and the taste and cooking properties differ greatly. Countries and regions have different cultural requirements for rice categories. In China, coastal people like hard rice with high AC, while inland people like soft rice with low AC. Therefore, it is necessary to develop multiple rice varieties.

Rice waxy gene *Wx* (LOC_Os06g04200), located on the short arm of chromosome 6, encodes granule-bound starch synthase (GBSS) and consists of 13 exons and 14 introns for a total length of 6185 bp. It is a major gene that controls amylose synthesis and directly affects amylose content (AC) in rice endosperm and pollen [6,7]. For non-glutinous rice varieties, the *Wx* gene is differentiated into two alleles, *Wx^a* and *Wx^b*. All wild rice is *Wx^a*; indica rice is mainly *Wx^a* with higher AC; almost all japonica rice varieties are *Wx^b* with lower AC [8,9], indicating that *Wx^b* evolved from *Wx^a*. The difference between the two alleles was mainly due to (CT)n polymorphism, which explained the AC difference between indica and japonica non-glutinous rice varieties [10]. Sequence analysis showed that compared to *Wx^a*, the mutation of GT→TT occurred at the 5′ end of intron 1 of *Wx^b*, which led to a decrease in splicing efficiency and abnormal splicing. The decrease in mature *Wx^b* transcript content and β-glucuronidase activity led to a decrease in amylose content [11,12]. Therefore, the regulation of AC is related to the ability to excise introns and belongs to post-transcriptional regulation. The genotype of waxy rice was *wx*, which produced amylopectin almost exclusively. The structure changes of the *Wx* exon or intron affected the gene expression and protein function, which greatly changed the AC [13]. Non-waxy gene (*Wx*) was not completely dominant over waxy gene (*wx*), and there was a significant dose effect [9].

Glutinous rice is a mutant type. Its endosperm appears milky-white, contains more than 95% amylopectin and little or no amylose. It has a special value in food, medicine, industry, and other fields. China's glutinous rice resources are quite abundant, but most cannot be directly used for production. At present, there are only 32 approved hybrid glutinous rice varieties in China, of which 28 are indica, and 4 are japonica, of which there are only 4 waxy PTGMS lines (https://ricedata.cn/) (accessed on: 15 September 2021). As we know, the key to developing two-line hybrid rice is PTGMS, and good PTGMS can be used to assemble a batch of excellent two-line hybrid rice varieties. Therefore, enriching high-quality PTGMS glutinous rice germplasm resources and quickly breeding new hybrid glutinous rice is a challenging goal.

The CRISPR/Cas9 system represents a new generation of gene editing that has immeasurable potential in basic research and application, especially in crop breeding to generate new germplasm resources rapidly. For ideal plant type, yield, grain quality, and resistance, the CRISPR/Cas9 system has shown amazing applications. Japanese scholars used it to knock out the *Wx* gene in the rice variety "Nipponbare" to breed a new glutinous variety in line with the tastes of the Japanese population [14]. Many studies have reported the use of gene editing to study yield-related quantitative trait loci (QTL), such as *CKX* [15], *Gn1a*, *IPA1*, *GS3*, and *DEP1* [16]. Researchers successfully applied CRISPR/Cas9 to obtain mutations in the *SPP*, *YSA*, and *ROC5* rice genes [17]. Zhang et al. [18] studied 11 target genes in two rice subspecies and confirmed the efficient application of the system in rice. Liu constructed a plant multi-target intelligent CRISPR/Cas9 targeting system [19], which further promoted its application in improving rice and other crops. By using the system, a set of mutants with important breeding values was obtained by site editing *TGW6*, a gene regulating 1000 grain-weight rice [20]. More importantly, CRISPR/Cas9-edited plants can remove express exogenous transgenes.

In 2018, the European Court of Justice ruled that gene editing crops belong to genetically modified crops (GMO) and follow the existing European GMO regulations. In particular, this means that gene editing crops should be subject to the same degree of regulation. However, the current mainstream view is optimistic about the development of gene-editing crops [21]. For example, Golden Rice passes the final regulatory hurdle in the Philippines. Farming can now start planting. Therefore, we believe that the new gene-editing germplasm waxy Y58S obtained through this study will play an important role in the foreseeable future. Previous studies have shown that the use of the CRISPR/Cas9 system can cause *Wx* gene mutation, reduce the amylose content of rice, and even obtain glutinous rice varieties [22,23]. However, there is no report on editing the *Wx* gene by CRISPR/Cas9 system to obtain the PTGMS line. Traditional breeding, mutation breeding and radiation breeding are the main methods for breeding glutinous PTGMS rice, but they are time-consuming and laborious. For excellent PTGMS lines and restorer lines, waxy PTGMS lines can be produced by editing the *Wx* gene mediated by CRISPR/Cas9, which will be a simple and efficient new method for breeding glutinous hybrid rice with high yield and quality

Y58S is the backbone of two-line hybrid rice in China. At present, more than 120 strong combinations of Y two excellent series have been selected. Forty-one of them have passed provincial-level certification, four have passed national-level certification, three have been identified for super rice. In terms of rice quality, two national-level combinations have reached the nationally approved second-grade quality rice, and nearly ten combinations have reached the third-grade high-quality rice standard. These varieties have been widely promoted in production with significant characteristics such as broad adaptability, high quality, multiple resistance, and ultra-high yield. At present, the cumulative promotion area has exceeded 2.7 million ha, an increase of 2 billion kilograms of grain, and economic benefits of more than RMB 4 billion (Https://ricedata.cn/) (accessed on: 15 September 2021).

In this study, the CRISPR/Cas9 system was used to effectively edit the high-quality PTGMS line rice Y58S, which is widely used in production, and cultivated waxy PTGMS line wx-Y58S with extremely low amylose content. In addition, wx-Y58S has maintained the advantage of being wild type and does not contain exogenous genetically modified ingredients. It is a new high-quality glutinous rice resource, which will have great application value in production. This study provides a simple and effective strategy for the breeding of glutinous two-line hybrid rice, which is of great significance to the cultivation and application of waxy two-line hybrid rice. In China, Y58S is a large-scale two-line male sterile line. Its combination has great heterosis, such as yield, resistance, etc. As long as the gene-editing crops can be supervised, this waxy-Y58S will be able to produce super waxy hybrid rice, which greatly improves glutinous rice yield and increase social benefits.

## 2. Materials and Methods

### 2.1. Experimental Material, Rice Material Planting, and Growth Environment

Y58S is an excellent PTGMS strain. A large number of excellent two-line rice hybrids and super rice varieties have been assembled by Y58S. Moreover, it has a large-scale planting area in China. With reference to previous studies, a CRISPR/Cas9 targeting vector was constructed for editing the expression vector [17]. Furthermore, co-culture of Y58S callus and *agrobacterium tumefaciens*-mediated transformation of EHA105 was carried out to obtain transformed plants [24]. In the normal growing season of rice, the transformed rice is planted in a mesh room with isolation conditions in Nanning, Guangxi. All materials were treated according to the conventional planting management method. After the male sterility was identified, the upper half of the plant was cut off and regenerated under cold water treatment at 22 °C day/19 °C night. After maturity, $T_1$ and $T_2$ seeds were obtained.

*E. coli* DH5α, *agrobacterium* EHA105, and Y58S rice materials, required for the experiment, were provided by the State Key Laboratory of Conservation and Utilization of Subtropical Agri-bioresources, Guangxi University. Required vectors and promoters:

pYLCRISPR/Cas9Pubi-H, pYLsgRNA-LzU6a, and U6a were provided by the State Key Laboratory of Conservation and Utilization of Subtropical Agri-bioresources, South China Agricultural University.

*Agrobacterium* EHA105 mediated the transformation of rice material Y58S callus, and after two screenings by adding hygromycin antibiotic medium, positive callus was obtained, and then, the seedlings were differentiated and the rooting was performed. Genetic transformation work was entrusted to Wuhan Boyuan Biological Co., Ltd. (Wuhan, China).

Experimental reagents include KOD-Plus, T4 ligase, Bas I-HF, Kanamycin, Ampicillin, MluI, plasmid mini-extraction kit, and PCR purification kit.

Primer synthesis and Sanger sequencing analysis were performed by Huada Biotechnology Co., Ltd. (Beijing, China) (primer information is shown in Table 1).

**Table 1.** Primer sequences used in this study.

| Primer Name | Primer Sequence 5'-3' |
|---|---|
| Wx-text-F | TCCGCCACGGGTTCCAG |
| Wx-text-R | CTCCTACCTCAGCCACAACG |
| U-F | CTCCGTTTTACCTGTGGAATCG |
| gR-R | CGGAGGAAAATTCCATCCAC |
| Wx-U6a-1F | TGTGTGCTTACAGCCATGGCGTTTTAGAGCTAGAAAT |
| Wx-U6a-1R | GCCATGGCTGTAAGCACACACGGCAGCCAAGCCAGCA |
| Pps-GGL | TTCAGAGGTCTCTCTCG<u>ACTAGT</u>ATGGAATCGGCAGCAAAGG |
| Pgs-GGR | AGCGTGGGTCTCGACCG<u>ACGCGT</u>ATCCATCCACTCCAAGCTC |
| PB-R | GCGCGCGGTCTCTACCGACGCGTATCC |
| PB-L | GCGCGCgGTCTCGCTCGACTAGTATGG |
| HPT-F | ATTTGTGTACGCCCGACAGT |
| HPT-R | GTGCTTGACATTGGGGAGTT |
| CAS9-F | CTGACGCTAACCTCGACAAG |
| CAS-R | CCGATCTAGTAACATAGATGACACC |

Note: <u>ACTAGT</u> and <u>ACGCGT</u> is *Spe* I and *Mlu* I sites.

### 2.2. Construction of pYLCRISPR/Cas9-Wx Vector

2.2.1. Target Site Selection and Vector Construction

According to the *Wx* (LOC_Os06g04200) sequence provided by the rice genome annotation website (http://rice.plantbiology.msu.edu/) (accessed on: 15 September 2021), the amplification of the *Wx* gene was performed for Taichung 65 and Y58S by using specific primers (Wx-text-F/R). The *Wx* gene is located on chromosome 6 and has a total length of 5032 bp. This reference sequence was used to design oligonucleotide primers for the amplification of *Wx* target positions. After confirming the gene sequence, two 20 bp-long sgRNAs sequences followed by PAM (protospacer adjacent motif) were designed according to the A/G(N) 20NGG in the first and second exon regions by using the CRISPR-P (cbi.hzau.edu.cn/crispr/) (accessed on: 15 September 2021) online tool (Figure 1A,B). The designed sgRNAs sequences were blasted against the rice genome using NCBI (https://blast.ncbi.nlm.nih.gov/Blast.cgi) (accessed on: 15 September 2021) to exclude non-specific target sites and confirm specificity. pYLCRISPR/Cas-PubiH vector and promoters (U6a and U6b) were obtained from Yaoguang Liu's lab, and the Oligo sequences and detection primers were synthesized by Beijing Genomics Institute, China.

2.2.2. Construction of pYLCRISPR/Cas9-Wx Vector

An expression cassette was generated according to Ma et al. [19], with slight modifications. The corresponding adapter primers Wx-U6-F/Wx-U6-R were used to construct the ligation reaction of the sgRNA expression cassette. Then, the ligation product was used as a template for PCR amplification. The expression cassette product was purified. Fifteen microliters of restriction digestion ligation reaction was prepared, including pYLCRISPR/Cas9 plasmid 60 ng, 10 ng sgRNA-osU6a expression cassette mixture, 10 × CutSmart Buffer, 10 mM ATP, 10U Bsa I-HF, 35UT4 DNA ligase, and variable temper-

ature cycling for 15 cycles of digestion. The vector fragments were ligated with T4 DNA ligase (NEB) at 20 °C for 2 h. After the ligation, the product was transformed into DH5α competent cells by heat shock method. The transformation product was cultured overnight, and a single clone was selected for inoculation and culture. After the plasmid extraction, a colony PCR was performed to assure the correct clone was sent for sequencing.

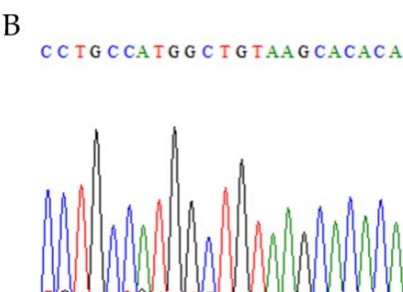

**Figure 1.** Location single guiding RNA (sgRNA) in the *Wx* gene and designed primers for amplification. (**A**) *Wx* gene partial sequence showing target positions and primer sequences; red nucleotides represent the upstream and downstream primers; black underlined nucleotides denote the protospacer adjacent motif (PAM) region; blue nucleotides show sgRNA sequences. (**B**) Sequencing peak map of sgRNA.

### 2.3. Genotyping, Phenotyping, and Screening of T-DNA-Free Plants

The target sites of the $T_0$, $T_1$, and $T_2$ generations of the genetic transformation material were sequenced and analyzed, and the PCR products were sequenced by the Sanger method. The CTAB method [25] was used to extract the genomic DNA from the fresh leaves of the mutant plants was extracted, and specific primers (Wx-test-F/R) were designed to detect the on-target mutations. The agronomic traits such as plant height, effective panicles, and grains per panicle, thousand-grain weight, grain length, grain width, and seed setting rate of Y58S and mutant lines were recorded in $T_0$, $T_1$, and $T_2$ generations. The genomic DNA of transformed plants was extracted from young leaves by the CTAB method for PCR amplification. The Cas9 gene-specific primers Cas9-F/Cas9-R were used to PCR-amplify the genomic DNA of mutant plants. PCR amplification conditions were as follows: initial denaturation at 95 °C for 5 min; followed by 35 cycles of denaturation at 95 °C for 30 s, annealing at 56 °C for 30 s, and extension at 72 °C for 30 s; followed by a final extension at 72 °C for 5 min. The PCR products were detected by agarose gel electrophoresis. The plants that failed to amplify were considered T-DNA-free.

### 2.4. Grain Phenotype of Endosperm and Iodine Staining

Mature grains of WT and mutant plants were taken and de-husked, and their appearance was observed. The endosperm about 1/3 from the embryo tip was excised, and 1% iodine reagent was dripped on the exposed surface of the endosperm and observed and photographed after 1 min. The grains surface of non-glutinous rice endosperm quickly turned blue-black, but the color of glutinous rice did not change (red-brown).

### 2.5. Determination of Amylose Content by Dual-Wavelength Method

Five selected $T_1$-generation T-DNA-Free plant and Y58S seeds were germinated and sown, and the seedlings were grown in a cold-water pool at 20–22 °C during the seedling stage and maintained regularly. $T_2$ generation mature seeds were harvested, dried in an incubator at 35 °C for two days, and stored at room temperature for more than 3 months for AC determination. A dual-wavelength spectrophotometric method was established to determine the AC in $T_2$ plants [26,27]. The optimal measurement and reference wavelengths of amylose were determined, according to the dual-wavelength isometric point method, and they were substituted into the regression equation to determine the AC in the sample.

### 2.6. Pollen Fertility Survey

Iodine staining of rice pollen is divided into two types: sterile pollen is mainly brown and yellow stained with 1% KI-$I_2$, 3 cases of typical abortion, round abortion, and dyeing abortion; fertile pollen is dyed black. Y58S and wx-Y58S were planted in cold-water pools with temperatures between 20–22° and summer fields in Nanning. During the rice flowering season, five plants of the above flowering rice (three spikelets per plant) were randomly selected and stained with iodine using 1% KI-$I_2$. The pollen grain color change was observed under the microscope.

### 2.7. Data Analysis

Data were processed using Excel 2016 version (Redmond, WS, USA), combined with IBM SPSS 20.0 software (Chicago, IL, USA) for statistical analysis, and single-factor analysis using DUCAN method to detect significant differences; $p = 0.05$ as the level of significant differences.

## 3. Results

### 3.1. Vector Construction

The construction fragment of a target sgRNA expression cassette of the *Wx* gene corresponds to a U6a-sgRNA amplification product that was 629 bp in length (Figure 2A). The constructed Cas9/sgRNA expression vector was transformed, cultured, and a single colony was selected. The universal primer UF and the target forward and reverse primers were used for PCR amplification and sequencing. The gel electrophoresis results showed that the *Wx* gene Cas9/sgRNA expression vector was present in all selected single colonies except WT1-8 (Figure 2B). Sequencing results confirmed that the target was present in the vector (Figure 2C).

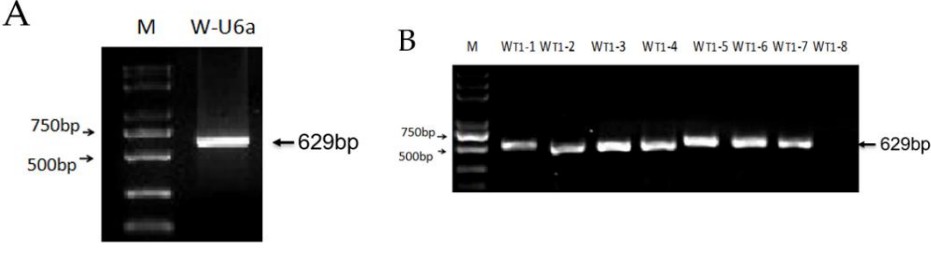

CGGGGTAGGATGCAATAGAGAGCAACGTTTAGTACCACCTCGCTTAGCTAGAGCAAAC
TGGACTGCCTTATATGCGCGGGTGCTGGCTTGGCTGCCGTGTGTGCTTACAGCCATGGC
GTTTTAGAGCTAGAAATAGCAAGTTAAAATAAGGGTAGTCCGTTATCAACTTGAAAAA
GTGGCACCGAGTCGGTGCTTTTTTTCAAGAGCTTGGAGTGGATGGAAAGGACTACATG

**Figure 2.** Construction of vector and verification of sgRNA. (**A**) Detection results of sgRNA expression cassette; (**B**) verification of the size of *Wx* gene cas9/sgRNA expression vector segment; (**C**) verification of *Wx* gene cas9/sgRNA expression vector sequencing. M: marker; WT1-1—WT1-8: $W_X$ gene cas9/sgRNA expression vector; 750 bp, 500 bp, 629 bp: fragment size; red part: target sequence.

### 3.2. Genotyping and Analysis of Mutation Types in $T_0$ Plants

A total of 28 $T_0$ plants were obtained in the $T_0$ generation, among them, 18 were mutant plants. The amplification results of the target fragments of some $T_0$ plants are shown in Figure 3. The mutation rate was 64.29%, indicating higher mutation efficiency.

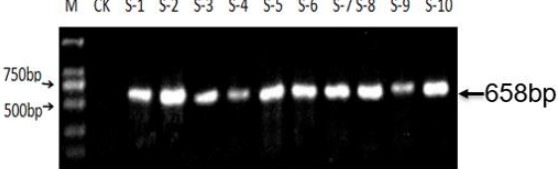

**Figure 3.** Amplification results of partial $T_0$ generation transformed plants of Y58S material. M: marker; CK: negative control; S1–S10: part of $T_0$ transformation plants; 750 bp, 500 bp, 658 bp: fragment size.

The genotype analysis of target mutations in the $T_0$ generation showed that most of the mutations were biallelic, whereas homozygous mutations were lower but there were not any chimeras. The homozygous mutation rate was 33.3% (Table 2), indicating that this target has a high editing efficiency and is a reasonable target for editing the Wx site.

**Table 2.** Mutation types in $T_0$ generation.

| Types | Heterozygous | Bi-Allelic | Homozygous | Total |
|---|---|---|---|---|
| Number | 5 | 7 | 6 | 18 |

Most of the mutations were insertions and deletions only. Mutations occurred in the first few bases of NGG (APM). The types of insertion mutations are mostly A/T base insertions. W-1 showed a replacement of 3 bases and 1 base insertion (G to T); W-4 and W-6 showed an insertion of 1 base, but not at the same site; W-13 and W-15 showed deletion of 5 bases and 1base, respectively (Figure 4, Table 3).

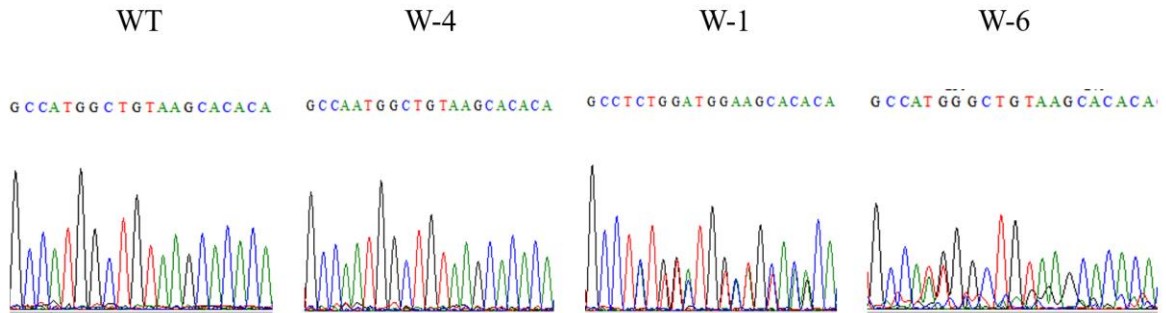

**Figure 4.** Sequencing results of main $T_0$ transgenic plants. WT: wild type(Y58S); W-1&W-4&W-6: mutant plants.

**Table 3.** Main mutation types of *Wx* gene target in Y58S $T_0$ transformation material of rice. WT: wild type; W-1 and W-4 and W-6: mutant type.

| Transformation Material | Mutation | Type |
|---|---|---|
| W-T | GCCA_TGGCTGTAAGCACACA | |
| W-1 | GCCTCTGGATGGAAGCACACA | +1/+3substitution |
| W-4 | GCCAATGGCTGTAAGCACACA | +1 |
| W-6 | GCCA_TGGGCTGTAAGCACACA | +1 |

### 3.3. Screening of T-DNA-Free Plants

The homozygous mutant plant W-4 of the $T_0$ generation was selfed to obtain the $T_1$ generation, and then, 5 $T_1$ plants H1-H5 were randomly selected for testing (Figure 5). The result showed that H-3 and H-5 do not carry exogenous T-DNA fragments, and both showed heritable homozygous mutations with the same pattern. The bi-allelic mutant line W-1 was also selfed, and 10 $T_1$ generation plants B1-B10 were obtained for testing (Figure 5). As a result, B-2, B-3. B-4, B-5, B-6, B-8, and B-10 did not carry exogenous T-DNA fragments, and the isolated plants B-2, B-5, and B-10 were homozygous by sequencing. At

this point, a total of five T-DNA-free homozygous plants were obtained. These plants were grown under natural conditions, and $T_2$ generations were obtained.

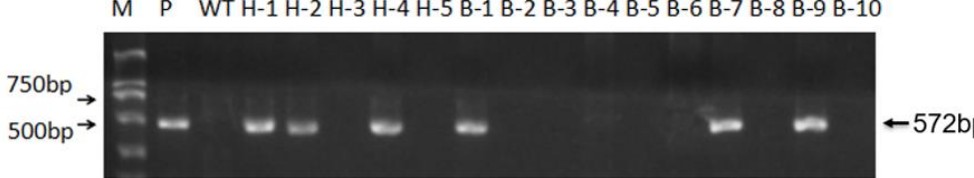

**Figure 5.** Test results of T-DNA-Free transformation of $T_1$ rice material Y58S. M: marker; P: positive control; WT: wild type; H-1–H-5: $T_1$ plants of $T_0$ W-4 strain; B-1–B-10: $T_1$ plants of $T_0$, W-1 strain; 750 bp, 500 bp, 572 bp: fragment size.

*3.4. Determination of Amylose Content in $T_2$ Generation*

The amylose content (AC) was determined according to the dual-wavelength isometric point method by drawing the reference wavelength analysis curve for AC and amylopectin (Figure 6A) [28] and the standard curve for amylose (Figure 6B). The AC measurement was adjected at a wavelength of $\lambda1 = 620$ nm and a reference wavelength of $\lambda2 = 469$ nm; and the regression equation $Y = 0.0013x + 0.0029$, $R = 0.9936$ of the standard curves was used to substitute the measured sample value. The AC concentration in the range of 10~60 mg/L was linear, and the AC in the sample was measured and calculated.

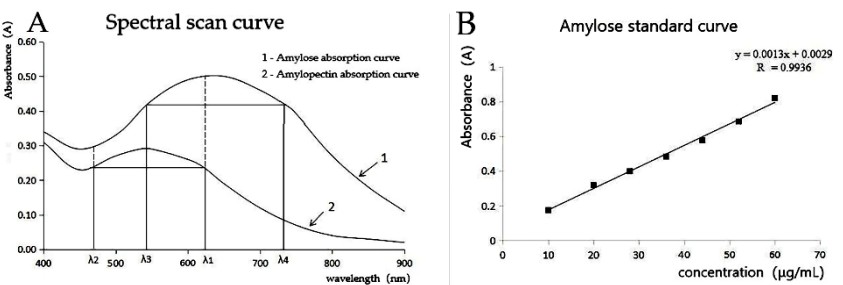

**Figure 6.** Drawing standard curve of AC. (**A**) The preparation analysis of amylose and amylopectin; (**B**) standard curve of AC.

The analysis results showed that the rice material Y58S AC was 13.8%. Homozygous $T_2$ mutant plants showed AC below than 2%, showing a strong waxiness, especially W-4-H-3, W-1-B-5, W-1-B-10, showed AC below 1%.

W-1-B-5 with the lowest AC in $T_2$ generation was selected for subsequent experiments and named wx-Y58S. Comparing the waxy appearance of the mutant and the WT, it was found that the endosperm of wx-Y58S showed a clear milky white color, which was significantly different from the endosperm color of Y58S, and the AC reached 0.6% (Table 4). After further endosperm iodine staining, it was found that the iodine staining the color of the cross-section of the endosperm of Y58S and wx-Y58S was black-blue and brown-red, respectively, which can significantly distinguish the waxiness of both (Figure 7).

**Table 4.** AC contents of polished rice of $T_2$ homozygous mutant lines.

| $T_2$ Line | AC (%) |
|---|---|
| W-4-H-3 | 0.8 |
| W-4-H-5 | 1.0 |
| W-1-B-2 | 1.8 |
| W-1-B-5 | 0.6 |
| W-1-B-10 | 0.9 |
| Y58S (control) | 13.8 |

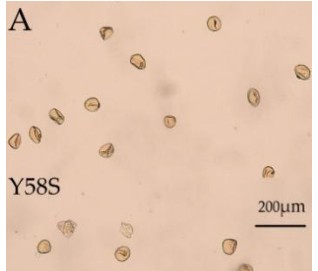
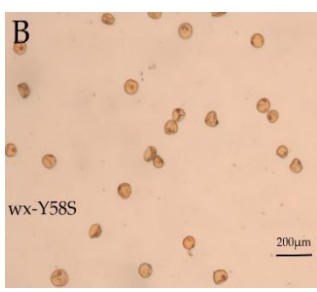
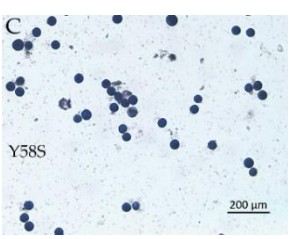
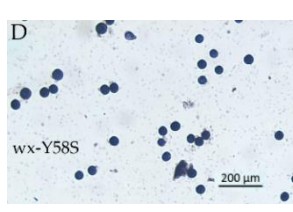

**Figure 7.** Pollen iodine staining results of Y58S (**A,C**) and wx-Y58S (T$_2$ line W-1-B-5) (**B,D**). (**A,B**) the normal growth summery condition of rice pollen in Nanning, Guangxi, showed round abortion sterility; (**C,D**) the fertility of rice pollen changed to fertile under the condition of cold pool 20–22° (bar = 200 μm).

Compared with the mutations of the T$_0$ generation type, the homozygous mutation performance of the waxy locus was stably inherited, and the number of plants is sufficient for subsequent use in breeding research work. The above shows that the W-4-H-3 line has been successfully transformed into glutinous rice wx-Y58S.

### 3.5. Plant Type, Fertility Transformation, and Appearance Analysis of Grain and Rice Kernels of T$_2$ Homozygous Mutant Lines

T$_2$-generation wx-Y58S and Y58S rice materials were planted in the same under controlled temperature (20–22 °C) and managed regularly. At the maturity stage, seven main agronomic traits of wx-Y58S strain and Y58S rice were investigated. The results revealed that the traits of the wx-Y58S lines were not significantly different from Y58S. The results indicate that the mutations in *Wx* genes do not change other agronomic traits (Table 5, Figure 8).

**Table 5.** Agronomic characters of wx-Y58S T-DNA-free T$_2$ homozygous mutants.

| T$_2$ Homozygous Mutation Lines | Plant Height (cm) | No. of Panicle/Plant | Flag Leaf Length (cm) | Flag Leaf Width (cm) | Length of Panicle (cm) | Grain no. per Spike | Grain Set Rate (%) |
|---|---|---|---|---|---|---|---|
| W-4-H-3 | 83.3a | 6 | 45.3 a | 1.5 a | 25.0 a | 183 a | 84.6 a |
| W-4-H-5 | 83.5 a | 6 | 45.5 a | 1.5 a | 26.2 a | 184 a | 85.0 a |
| W-1-B-2 | 82.7 a | 6 | 46.2 a | 1.6 a | 25.8 a | 181 a | 85.2 a |
| W-1-B-5 | 81.9 a | 7 | 46.5 a | 1.5 a | 25.8 a | 180 a | 85.0 a |
| Y58S (CK) | 82.7 a | 7 | 46.8 a | 1.6 a | 26.2 a | 186 a | 85.5 a |

Note: a, $p = 0.05$ level, no significant difference.

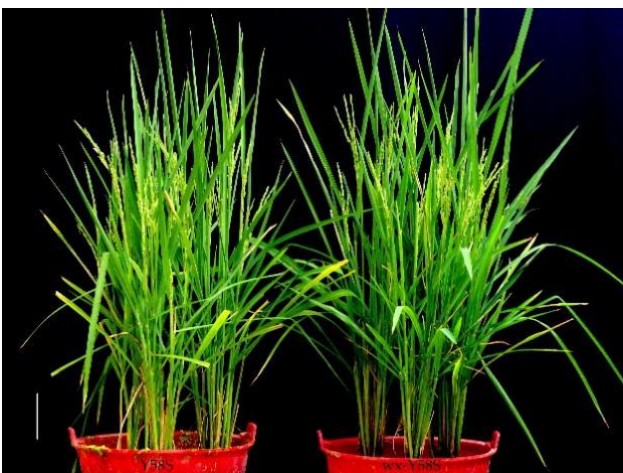

**Figure 8.** Plant type comparison of wx-Y58S (T$_2$ W-1-B-5 plants) (left) and the control Y58S plant (right) (bar= 10 cm).

The selfing seeds of wx-Y58S strain and Y58S in the cold-water pool were harvested, respectively. Their grain appearance, polished rice appearance, and iodine staining results of polished rice endosperm cross-section were observed and compared. It was found that the grain appearance quality such as grain length and grain width of wx-Y58S strain was not significantly different from Y58S. However, the difference between the appearance color of polished rice and the iodine-stained color of the endosperm cross-section is obvious, which can clearly distinguish the waxy difference (Figure 9A–C). It is shown that wx-Y58S changes the waxiness of Y58S without affecting the appearance of grains and rice grains.

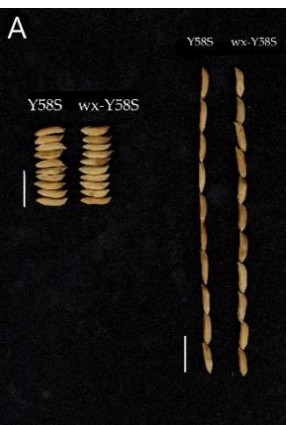 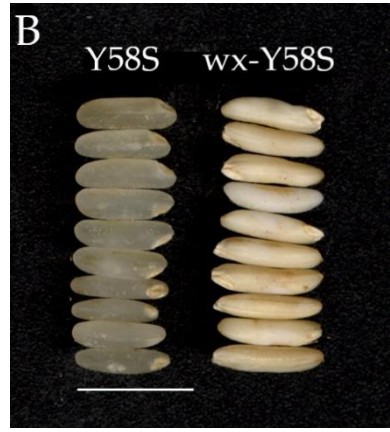 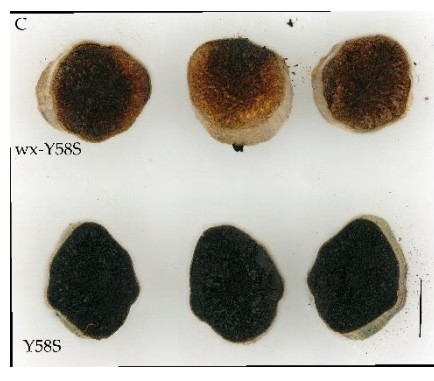

**Figure 9.** Comparison of rice grain phenotypes of Y58S and wx-Y58S (T$_2$ line W-1-B-5). (**A**) Grain; (**B**) brown rice; (**C**) iodine-stained endosperm cross sectioning. (**A**,**B**): bars = 1 cm; (**C**): bar = 1 mm).

In the normal rice-growing season, T-DNA-free wx-Y58S and Y58S rice materials were planted in a Nanning field and a cold-water pool temperature-controlled at 20–22 °C. At the heading stage, pollen iodine staining was performed to investigate fertility. The results showed that the pollen-iodine staining of wx-Y58S and Y58S plants that were normally grown in the field was reddish-brown, that is, they were round sterile; while the pollen of wx-Y58S and Y58S plants that were normally grown in cold water Iodine staining was all black-blue, meaning fertility was transformed into fertile (Figure 7A–D). In the later stage, it was also found that the sterile plants in the field did not bear seeds, and the fertile plants in the cold-water pool gave birth to seeds. These results indicated that the mutation sites of these mutant lines have no significant effect on the fertility characteristics of rice plants.

*3.6. Analysis of the Genetic Background of Y58S*

The molecular markers of 50 indica-japonica comparisons distributed throughout the genome were used to analyze the indica-japonica background of Y58S [29]. It was found that 15 markers were consistent with the Japonica rice variety Nipponbare. Except for the fact that no markers were detected on chromosomes 1 and 8, other chromosomes were detected, and they were mainly concentrated on chromosomes 4 and 5. The remaining 35 markers were consistent with indica rice variety 9311. These results indicate that Y58S does belong to the indica sterile line but also contains 1/3 of the japonica rice background, which is consistent with the analysis of the pedigree of Y58S, which may be derived from its modern parents, such as Lemont and Nongken 58. Therefore, wx-Y58S obtained by gene editing also contains part japonica components, which is a high-quality glutinous rice line.

## 4. Discussion

In the field of crop breeding, the improvement of individual characters by plant genome editing technology has a great advantage over the traditional breeding methods. Although the traditional backcross breeding method can achieve similar breeding purposes to a certain extent, it is difficult to obtain new improved varieties in breeding practice due to long cycles, heavy workloads, and sometimes cumbersome chain effects. Using

genome editing technology to modify the target sites can directly obtain new varieties modified by characteristic sites, which can not only shorten the breeding cycle and reduce the workload but also effectively avoid the linkage effect and improve the breeding success rate and breeding efficiency [30]. Furthermore, the innovation and improvement of genome editing technology provide a wide range of opportunities for plant breeding. We can use genome editing technology to improve crops with multiple targets and traits, including prolonging or shortening the growth period, enhancing or weakening photosensitivity, improving quality traits, improving plant height, and other plant type traits. The efficient CRISPR/cas9 plant genome multi-target editing system for monocotyledons and dicotyledons constructed by Professor Liu Yaoguang of South China Agricultural University has wide application and important influence in gene targeting and genome editing breeding [19]. Using this system, we have done a lot of work in the improvement of rice characters, including editing and modifying the genes of *SRL1,2* [31], *SD1* [32], *TGW6*, *Wx* [33], etc., and obtained a large number of breeding materials with good plant leaf shape and good rice quality. It also shows that CRISPR/cas9 system has high efficiency in rice genome editing.

CRISPR/cas9 system sometimes has off-target effects when editing the target site, which is also a limitation of CRISPR/cas9 system. However, it is undeniable that in the process of the CRISPR/cas9 system operation, with the continuous optimization of the system, these days the off rate is not so high. CRISPR/cas9 gene-editing technology was used to edit the genome to obtain mutants that can stably express and inherit the offspring [18]. In this study, most of the target site mutations were biallelic mutations, followed by homozygous mutations; target site biallelic mutations occurred, and the offspring were separated into homozygous mutants and biallelic mutants according to Mendelian genetic law; the homozygous mutations in the genetic separation without CRISPR/cas9 system, the offspring, were still homozygous mutations and could be inherited stably. In conclusion, CRISPR/cas9, as an effective gene-editing system at present, has broad application prospects in improving rice characters and breeding new varieties.

Glutinous rice is a favorite traditional national food and is widely used in medical, industrial, and other fields. With the improvement of people's living standards, the market demand for glutinous rice, especially high-quality glutinous rice, is increasing year by year [32]. At present, conventional rice is still the main glutinous rice in production, which has serious problems of low yield and poor resistance [34–36]. In recent years, some hybrid glutinous rice has been approved, which overcomes, to some extent, the problem of low yield, but the problem of poor quality is prominent. Breeding high-quality and high-yield hybrid glutinous rice and genetic improvement of new glutinous rice varieties may be effective ways to solve the market demand for high-quality glutinous rice in the future. Conventional waxy rice crossbreeding has a long time, low breeding efficiency, and low yield potential. Two-line hybrid rice is one of the main types of hybrid rice in China. It simplifies the breeding procedure, saves the cost of seed production, has the advantages of convenient combination and high yield, and has a large number of high-yield two-line hybrid rice varieties applied in production [37]. Through gene editing of *Wx* locus, the rapid transformation of high-yield and high-quality two-line hybrid rice into two-line glutinous rice varieties is a key measure to greatly improve the yield level of high-quality glutinous rice [22,23]. In this study, glutinous PTGMS line wx-Y58S was obtained by gene editing of Y58S, which was widely used in production. Except for its waxy character, the glutinous PTGMS line wx-Y58S was consistent with Y58S in other aspects (Figure 8, Table 5), which showed important utilization value and laid the foundation for cultivating high-yield waxy two-line hybrid rice. It is important that wx-Y58S be used to mix waxy hybrid rice with glutinous restorer lines. It is expected that it will be possible to select high-yield glutinous two-line hybrid rice varieties in the short term.

Glutinous rice, for a long time, has not been a staple food, so it has not received enough attention, and the development of glutinous rice breeding is slow. However, in recent years, due to the special nature of glutinous rice, the market demand for glutinous

rice has increased year by year. Glutinous rice production has become an indispensable industry in grain production. At present, the breeding level of glutinous rice in China is relatively low, and the main problems are lack of germplasm resources, low yield, poor resistance, and adaptability [38]. Glutinous PTGMS rice germplasm resources are the key to two-line hybrid glutinous rice. The excellent hybrid glutinous rice has wider adaptability than conventionality; the yield increases by 20–30% [31], and the effect of increasing production is very significant. Thus far, only four PTGMS varieties of glutinous rice have been approved in China (https://ricedata.cn/) (accessed on: 9 September 2021).

The quality of glutinous rice is closely related to the genetic background. The waxiness of Japonica glutinous rice is much better than that of indica, but there are no high-yielding two-line japonica hybrid rice varieties in production. Therefore, we aim to use indica japonica introgression lines and restorer lines with larger japonica background as gene editing materials to obtain hybrid rice with good glutinous properties. Through pedigree analysis (https://ricedata.cn/) (accessed on: 9 September 2021), it was found that Y58S was derived from Peiai 64S with tropical japonica rice, and lemon, one of the female parents, also had japonica genetic components [39,40]. Our study shows that Y58S has 1/3 japonica background, which is consistent with the results of pedigree analysis of Y58S, indicating that Y58S also has some japonica background (Figure 10A–C). Therefore, using Y58S as a *Wx* gene-editing material, we hope to obtain two-line male sterile lines with good waxiness. The results of this study (Table 4) showed that the amylose content of wx-Y58S modified by gene editing reached a very low level (as low as 0.6%), and the waxy quality was excellent. Y58s is a PTGMS line of super rice and an ideal material for *Wx* gene editing and improvement.

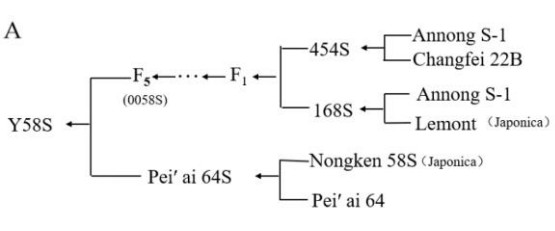

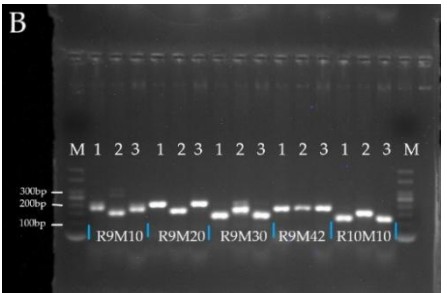

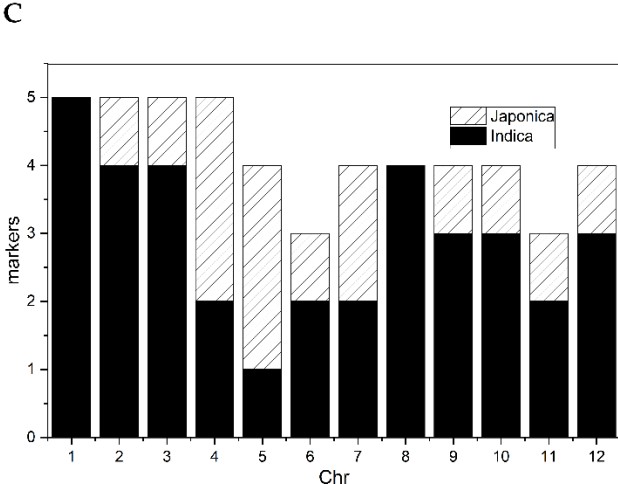

**Figure 10.** Pedigree and japonica background of Y58S. (**A**) Y58S pedigree; (**B**) Y58S background results of japonica rice detection. M: Marker; 1: 9311; 2: Nipponbare; 3: Y58S; R9M10R9M20R9M30, R9M42, R10M10: molecular detection markers; 100 bp, 200 bp, 300 bp: fragment size. (**C**) Distribution of Indica-japonica genetic background on 12 chromosomes in Y58S.

The structure change of the *Wx* exon or intron will affect the expression and protein function of the *Wx* gene, so we can choose a wide range of target gene sequences. The GC content of the target sequence has a great impact on the editing efficiency of the CRISPR/cas9 system [20]. When the GC% of the target sequence is 50–70%, it has a high shooting efficiency. There should be no more than four consecutive t in the target (5-N20NGG-3 direction) in order to prevent RNA Pol III from being taken as a transcription termination signal. In this study, we designed a target in the first exon of the Waxy gene, and the results showed that the CRISPR/Cas9 system can accurately and efficiently edit rice *Wx* genes. In addition, it is convenient to identify the genotype of the mutant by sequencing, and the mutant phenotype is also obvious. The CRISPR/Cas9 editing transfers PTGMS Y58S to waxy wx-Y58S without affecting other ideal agronomic traits and can successfully remove exogenous T-DNA fragments from the offspring. These works provide a simple and effective method for cultivating glutinous new PTGMS rice materials.

In this study, we only designed a target in the first exon of the *Wx* gene, which made the work simple and cheap. However, we obtained three important homozygous mutant lines W-1, W-4, and W-6, in which W-1 is a substitution and insertion mutation, while W-4 and w-6 are a base insertion. The AC of these three mutant lines decreased significantly, and W-1 and W-4 decreased below 1%. This result shows that the knockout of a target site can destroy the *Wx* gene so that it does not play a role and cannot synthesize amylose, which is consistent with previous studies: "the structural changes of exon or intron of *Wx* will affect gene expression and protein function, so as to greatly change the amylose content of rice" [13]. In addition, the agronomic characters of the three lines have not changed significantly, which is favorable information. It shows that we only changed AC and maintained the superiority of the original material, making it more conducive to production. It also shows that the CRISPR/cas9 gene-editing system has a low off-target rate, which once again confirms the great advantage of CRISPR/cas9 in crop character improvement.

Previous studies have shown that stable *Wx* mutants can be obtained through directional editing of the *Wx* gene by CRISPR/cas9, which can significantly reduce the amylose content (AC), which is a new waxy material [22,23]. In our study, the amylose content (AC) of Y58S was 13.8%, and the AC of the W-1-B-5 plant line we bred was as low as 0.6%. which completely met the AC standard of high-quality glutinous rice. It indicated that Y58S had been successfully transformed into glutinous rice, and we named it wx-Y58S. The results show that the editing effect of the target is excellent, and it is an ideal target for transforming non-glutinous rice into glutinous rice. At the same time, wx-Y58S maintained the excellent agronomic traits of wild-type Y58S and did not carry transgenic components. Y58s is the backbone parent of two-line hybrid rice in China and has a large popularization area, which indicates that wx-Y58S has greater research and application value.

## 5. Conclusions

This study reveals that the CRISPR/Cas9 system can generate a new glutinous PTGMS line without exogenous T-DNA components by editing the waxy genes of indica rice such as Y58S. By changing the *Wx* gene without affecting other agronomic traits, other sterile lines were integrated into glutinous sterile lines. As known, Y58s is a two-line parent widely used in production. The selected target site can greatly reduce the AC of rice, which is 0.6%. wx-Y58S, which we transferred, will also be a widely used waxy parent. This method provides a simple and effective strategy for breeding high-yield, high-quality waxy two-line hybrid rice, has great significance for large-scale application of waxy two-line hybrid rice, and provides excellent germplasm resources.

**Author Contributions:** K.T., X.W., and X.G. contributed equally to this work; K.T. was responsible for writing, revising, and publishing this article; X.W. was responsible for target design and vector construction; X.G. was responsible for the planting and identification of rice materials; Professor Y.L. provided relevant carriers; Professor R.L. was responsible for the overall conception and modification of the article. All authors have read and agreed to the published version of the manuscript.

**Funding:** This research was funded by the Guangxi Key R&D Project (Project Number: Guike AB16380066) and the Guangxi Innovation-Driven Development Special Fund Project (Guike AA17204070).

**Data Availability Statement:** The study did not report any data.

**Conflicts of Interest:** The authors declare no conflict of interest.

## Abbreviations

| CRISPR/Cas9 | Clustered regularly interspaced short palindromic repeats/Cas9 |
| AC | Amylose content |
| PAM | Protospacer adjacent motif |
| QTL | Quantitative trait loci |
| PTGMS | Photothermo-sensitive genic-male-sterile |
| WT | Wild type |
| Wx | Waxy |
| T-DNA | Transfer DNA |

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
