# Peer review of "Generation of a New Glutinous Photothermosensitive Genic-Male-Sterile (PTGMS) Line by CRISPR/Cas9-Directed Mutagenesis of Wx in Rice (Oryza sativa L.)"

_agriculture, doi:10.3390/agriculture11111044_

Round 1

Reviewer 1 Report

  • Abstract: need one or sentence about the future study of results of this study!
  • Introduction: The two references, Wang et al. (6) and Wang et al. (7) are rather old to support the authors' claim that a major gene that controls amylose synthesis and directly affects amylose content (AC) in rice endosperm and pollen, especially first reference. Consider adding a more recent reference if available.
  • Sano (8), Wang et al (9), and Sano (10) are more than thirty-year-old references, they claimed " gene was differentiated into two alleles" Is there any change in this classify?
  • Aims of this study are not clear?
  • Many references are not in order, especially in the materials and methods section!
  • The experimental design was not clear! What kind of experimental design was used?
  • Table 5: There are STATISTICAL differences among agronomic characteristics, but they are not SIGNIFICANT differences! Is there any interpretation for that?
  • The discussion section was lacking in discuss a relationship among the results, and it wrote as a review of results!
  • Missing conclusion to this manuscript!
  • References should be following journal style!
  • Other comments in the attachment, please note.

Author Response

  1. Dear reviewer, thank you very much for your reply. It's a great honor to answer your question. We have carefully revised the manuscript according to your comments.
  2. Future research prospects: it can be predicted that Waxy Hybrid Rice will have a huge application space in the foreseeable future.
  3. Reference (6)Wang et al. (1990)was updated to Zeng et al.(2020).
  4. Sano(8), Wang et al.(9), Sano(10),  they claimed " gene was differentiated into two alleles. Up to now, more than 10 rare multiple alleles have been identified in Wx gene, but among non glutinous rice varieties, indica rice is mainly wxa and japonica rice is basically WXb. References (8) and (10) are the same, which is our mistake. Now (10) is revised to Xu et al.(2021).
  5. The purpose of this study is to damage the function of Wx gene by gene knockout, reduce the amylose content in rice endosperm, and obtain a new waxy rice line with very low or even no AC, which may be used in large-scale production in the future.
  6. The references are rearranged according to the requirements of journal.
  7. Randomized block design was used in the experimental design with three biological replicates to determine the amylose content and other related indexes
  8. Table 5: There are STATISTICAL differences among agronomic characteristics, but they are not SIGNIFICANT differences! Is there any interpretation for that? It may be that the target site is located in the Wx gene, and the gene knockout of this site does not cause the changes of relevant genes related to agronomic traits, such as plant height, tiller number, grain weight and so on, so there is no significant difference in agronomic traits. This is the advantage of gene editing, which changes only a single trait and remains unchanged as a whole.
  9. The conclusion of this manuscript is that new germplasm without foreign gene expression can be quickly obtained through single target editing of Wx gene
  10. We have carefully revised the manuscript according to the comments in the annex

Reviewer 2 Report

Dear Authors,

I have only minor comments on the manuscript.

In the introduction, the sentence found in lines 106-108 is not true. Please check how the European Union approaches genome editing and correct this statement.

The length of the Wx gene found in the introduction and in the materials and methods are different, why?

Figure 6 is unreadable, it needs to be larger.

Best regards,

M.

Author Response

Dear reviewer, thank you very much for your suggestion. We are very happy to revise the manuscript according to your opinion.

  1. We revised the manuscript according to the EU's management of gene editing crops, as follows: In 2018, the European Court of Justice ruled that gene editing crops belong to genetically modified crops (GMO) and follow the existing European GMO regulations. In particular, this means that gene editing crops should be subject to the same degree of regulation. However, the current mainstream view is optimistic about the development of gene editing crops. For example, Golden Rice passes the final regulatory hurdle in the Philippines. Farming can now start planting.
  2. Because we are concerned about the correctness of the target sequence, and these two sequences are part of the Wx gene, and we did not intercept the same length. But it is actually consistent, which is the length of the intercepted display is different.
  3. We modified the clarity of Figure 6 so that it can now be seen clearly.

Reviewer 3 Report

The manuscript “Generation of a New Glutinous Photothermo-sensitive Genic-Male-Sterile (PTGMS) Line by CRISPR/Cas9‑directed Mutagenesis of Wx in Rice (Oryza sativa L.)” by Teng et al., is based on the effective use of the CRISPR / Cas9 system to mediate gene editing of the Wx gene of rice PTGMS line Y58S. The research objectives were achieved through inducing mutations in ultra-low amylose content (AC) to obtain PTGMS glutinous rice with excellent waxiness strain. This study can be valuable in the agriculture sector to provide excellent sterile lines for the large-scale application of glutinous two-line hybrid rice.

The manuscript is well written and thoroughly described however, the language used in the manuscript is not up to the standard, and several major and minor mistakes are shown, and many of the phrases are not clear. It should go through complete proofreading by a native speaker. Few examples of language mistakes are mentioned here below

Title: Generation of A New Glutinous …. “A” should be in lower case

There should be space after “Genic-Male-Sterile” before the bracket

Oryza Sativa should be italicized

Keywords must be different from title words

Abstract:

Abstract summarized the work precisely. However, there are some minor mistakes

Line 13: a new sentence cannot start with “And” …. please check and correct

Line 19: Is To a control treatment? Please specify here

Line 26: what is meant by “extreme PTGMS” I think the word extreme is inappropriate

Also mention the future perspective of this study at the end of the abstract section, talking about how this study can benefit the researchers and farming community.

Introduction

The introduction is to the point, however, mention the agronomic importance of rice crop in this section, in order to relate with your achieved objectives and mention its production/yield-related information for your country. Moreover, the introduction section needs some attention regarding its organization for grammar and sentences errors which creates confusion.

The reference numbering cannot be superscript. Please follow the journal’s format

At the end of the introduction section, also state that how this research can benefit the community in coming future.

Materials and Methods

The methodology of the proposed study is well written and properly described. However, I feel the picture quality in figure 1 is poor and it should be improved.

Heading 2.3: there is a repetition of the word “and” replace the first one with a comma

Procedures described in heading 2.4 needs reference if available. The same is the case with heading 2.6 as well.

Results and Discussion

The other major concern is the too lengthy discussion of results. Which is a good thing but on the other hand it makes it very exaggerated and speculative. The authors should mention solid and particular reasons for their arguments developed in this experiment. Moreover, there are some irrelevant and outdated citations in this section as well. This section needs critical revisions

The other major drawback is the language/grammar used in this section that is not up to the standard, which makes the discussions of results very speculative and draws down its significance.

A separate heading of conclusions should also be added

References

The references are not properly formatted. Please follow the format as per the author’s guidelines provided by the journal.

To sum up, this manuscript can find interest to readers in this field. However, I feel that the above-mentioned comments should be addressed.

Author Response

Dear reviewer, thank you very much for your comments. We are very glad to receive your encouragement. We have carefully revised the manuscript according to your comments.
Thank you and best wishes

  1. According to your opinion, we revised the manuscript and changed the key words to: rice; Genome editing; CRISPR/Cas9; Waxy; amylose content (AC); Y58S
  2. Line 26: what is meant by “extreme PTGMS” I think the word extreme is inappropriate. We removed the word "extreme".
  3. At the end of the summary, we supplement the prospect of this study: It can be predicted that Waxy Hybrid Rice will have a huge application space in the foreseeable future. Once put into use, it will greatly improve the yield of glutinous rice and increase social benefits.
  4. At the end of the introduction, we added some content about the future prospect of this study:In China, Y58S is a large-scale two-line male sterile line. Its combination has great heterosis, such as yield, resistance and so on. As long as the gene editing crops can be supervised, this waxy-Y58S will be able to produce super Waxy Hybrid Rice, which greatly improve glutinous rice yield and increase social benefits.
  5. The conclusion is added after the discussion
  6. The results and discussion are modified
  7. According to the requirements of periodicals, the references are rearranged to meet the requirements of periodicals

Reviewer 4 Report

It needs some English modifications

Do you have data for yield performance for transgenic comparing with the control especially for yield and grains quality?

Author Response

Dear reviewer, thank you very much for your comments. We are very glad to receive your encouragement. We have carefully improved the English of the manuscript according to your comments.
Thank you and best wishes

At present, we have not assembled a hybrid rice combination of waxy y58s, because the gene editing crop is still in the regulatory system. However, wild-type Y58S has a large application area in China. It is one of the parents of super hybrid rice and is famous for its high yield and high resistance. In particular, its combined Y Liangyou 2 and Y Liangyou 900 were the first to break through the targets of large area per unit yield of 13.5 t / hm2 and 15 t / hm2 respectively, achieved a continuous major breakthrough in the yield potential of super hybrid rice. Therefore, Y Liangyou No. 2 and Y Liangyou 900 were rated as China's top ten scientific and technological progress in 2011 and 2014 respectively by academicians of the two academies.